# Physico-Mechanical and Antibacterial Properties of PLA/TiO$_2$ Composite Materials Synthesized via Electrospinning and Solution Casting Processes

**Shiyi Feng [1,†], Feng Zhang [2,†], Saeed Ahmed [1] and Yaowen Liu [1,2,3,*]**

1   College of Food Science, Sichuan Agricultural University, Yaan 625014, China
2   School of Materials Science and Engineering, Southwest Jiaotong University, Chengdu 610031, China
3   California NanoSystems Institute, University of California, Los Angeles, CA 90095, USA
*   Correspondence: lyw@my.swjtu.edu.cn
†   These authors contributed equally to the work.

**Abstract:** In this study, PLA/TiO$_2$ composites materials were prepared via electrospinning and solution casting processes. By testing the mechanical properties, water contact angle, water vapor permeability, and solubility of the composite nanofibers and films, the comprehensive performances of the two types of nanocomposites were analyzed. The results show that maximum tensile strengths of 2.71 ± 0.11 MPa and 14.49 ± 0.13 MPa were achieved for the nanofibers and films at a TiO$_2$ content of 0.75 wt.%. Moreover, the addition of TiO$_2$ significantly cut down the water vapor transmittance rate of the nanofibers and films while significantly improving the water solubility. Further, the antibacterial activity increased under UV-A irradiation for a TiO$_2$ nanoparticle content of 0.75 wt.%, and the nanofiber and films exhibited inhibition zones of 4.86 ± 0.50 and 3.69 ± 0.40 mm for *E. coli*, and 5.98 ± 0.77 and 4.63 ± 0.45 mm for *S. aureus*, respectively. Overall, the performance of the nanofiber was better than that of the film. Nevertheless, both the nanocomposite membranes satisfied the requirements of food packaging materials.

**Keywords:** PLA; TiO$_2$; nanofibers; films

## 1. Introduction

The widespread use of synthetic polymers has led to many environmental problems, primarily due to their non-degradability. This phenomenon has stimulated interest in research on the production of natural polymer-based biodegradable packaging systems. Bio-based and biodegradable polymers have recently garnered attention as environmentally friendly alternatives to traditional plastic packaging materials owing to their capacity to lessen the environmental impact. Polylactic acid (PLA) has good biocompatibility, rigidness, and great thermoplastic for food packaging applications. However, PLA is highly brittle and has poor gas and moisture barrier properties [1–3].

Titanium dioxide (TiO$_2$) has been widely investigated over the past decade because of its antimicrobial property caused by photocatalysis under light that generates enough energy for the production of reactive oxygen species (ROS) such as •OH, H$_2$O$_2$, and O$_2$−•, which have the great potential in damaging microbial cells [4]. Moreover, under the ultraviolet irradiation, the food-borne microbes and allergens could be protected well [5]. Nanocomposite of composite TiO$_2$ have been comprehensively studied for different purposes such as amelioration in physical performances, enhancement in the biodegradability of synthetic polymers, active packaging with antibacterial activity for foodstuff, and degradation of organic pollutants. Alizadeh et al. prepared a whey protein isolate/cellulose nanofiber with TiO$_2$ and rosemary essential oil (REO) which might be a preservative

for mutton microbes and sensory qualities. They found that the nanocomposite films containing $TiO_2$ and REO inhibited the inhibit the growth of spoilage and pathogenic bacteria in meat, improved the organoleptic properties of meat samples, and increased the longevity of meat samples [6]. Gumiero et al. employed a blown film extrusion process to prepare a high-density polyethylene/calcium carbonate film that contained $TiO_2$, and researched the influences of photocatalytic activity on the structural and microbiological stability of a short-ripened cheese [7].

In recent years, much attention has been paid to PLA/$TiO_2$ nanocomposites. PLA/$TiO_2$ nanocomposites have good mechanical, thermal, and photocatalytic properties, and have potential applications in the fields of adsorption and photocatalytic removal of environmental pollutants, food preservation, coating materials, drug-controlled delivery systems, wound repair, and biosensors. Toniatto et al. developed nanostructured PLA/$TiO_2$ nanofibers by electrospinning. The nanostructured PLA/$TiO_2$ nanofibers presented no mammalian cell toxicity and, most importantly, possessed bactericidal activity with larger $TiO_2$ loadings [8]. Xing et al. investigated the effects on the antibacterial and physical properties of polyethylene (PE)-based films after being loaded with nano-$TiO_2$ particles; their conclusions indicate that the $TiO_2$ nanoparticles play the important role in antibacterial activity of PE/$TiO_2$ films [9]. Particularly, the microbicidal property of the PE/$TiO_2$ composite film was more impactful against *S. aureus*. Furthermore, they thought that this type of $TiO_2$-incorporated film might sustain a high exchange of water molecules throughout the packaging system, which would prevent high relative humidity within the packing, thus delaying fruit spoilage and increasing the shelf life of fruits and vegetables with higher respiratory rates. Therefore, this type of film is a good candidate for application as an active food packaging system [10].

In this study, PLA/$TiO_2$ composite films and nanofibers with PLA as the film-forming substrate and $TiO_2$ as the functional substance were prepared by solution casting and electrospinning methods, respectively. In the solution casting method, hybrid materials are prepared by direct dispersion of nanoparticles in a polymer matrix. On the other hand, electrospinning is of interest for surface coating of in situ generated nanoparticles specifically on the surface or in the bulk of a polymer matrix. To determine the differences between the nanocomposite membranes prepared by the two methods for application in antimicrobial packaging, the density, contact angle, water vapor permeability (WVP), light transmittance, opacity, mechanical properties, and antibacterial activity of the nanocomposites were studied to comprehensively analyze their practical application value.

## 2. Materials and Methods

### 2.1. Materials

An L-isomer consisting of 95.7% PLA was purchased from Nature Works LLC (Blair, NE, USA) $TiO_2$ nanoparticles (80% anatase and 20% rutile; diameter: 10 nm) were supplied by Evonik Degussa GmbH (Essen, Germany). Chloroform and all the other solvents were of reagent grade or higher purity, and were purchased from Chengdu Kelong Reagent Co. (Chengdu, China) unless otherwise indicated.

### 2.2. Electrospinning

A PLA solution (10%, *w/w*) was prepared by dissolving 1 g of PLA in a 10 mL co-solvent system with a DCM-to-DMF ratio of 7:3 (*v/v*) under constant stirring at 800 rpm for 6 h. A certain amount of $TiO_2$ was added into the homogeneous PLA solution to form the PLA/$TiO_2$ basic mixed solution. The polymer solution was extracted using a 5 mL syringe with a blunt metal needle as the nozzle. Control the distance between the collector and the tip to be maintained at around 15 cm, and a precision pump (Zhejiang University Medical Instrument, Hangzhou, China) was used to set the flow rate as 1.0 mL/h. By using a high-voltage statitron (Tianjing High Voltage Power Supply Company, Tianjing, China), the electrospinning voltage was controlled within 18 kV. A grounded roller collector was used to collect the fibers. Before use, the overnight vacuum-dried need to be done at room temperature of the fibers which aimed to eliminate any residual solution. The electrospun fibers were denoted as

PLA/TiO$_2$-1.5, PLA/TiO$_2$-1.25, PLA/TiO$_2$-1, PLA/TiO$_2$-0.75, and PLA/TiO$_2$-0.5, which correspond to PLA-to-TiO$_2$ weight ratios of 100:1.5, 100:1.25,100:1.0, 100:0.75, and 100:0.5, respectively.

## 2.3. Solution Casting

A certain amount of PLA particles was accurately weighed and dissolved in a chloroform solution, and magnetically stirred at 40 °C until complete dissolution into a transparent and uniformly viscous solution. Then, 2 wt.% of Span 80 was added as a thickening agent, and a polylactic acid membrane solution with 4 wt.% mass fraction was obtained. The same amounts of TiO$_2$ were dissolved in 20 mL of absolute ethanol to obtain five different nanocomposites with PLA-to-TiO$_2$ weight ratios of 100:1.5, 100:1.25, 100:1.0, 100:0.75, and 100:0.5. After ultrasonic dispersion for 60 min, the mixture was slowly added dropwise to the stirred PLA solution at a speed of two drops per second. After homogenization, ultrasonic dispersion was carried out for 30 min to obtain a composite membrane solution. Then, 45 g of different concentrations of the composite membrane liquid were respectively cast on glass plates with dimensions of 24 cm × 24 cm, and dried in a vacuum oven at 55 °C for 25 min to prepare the composite membranes. Finally, the membranes were dried in air for 48 h until the chloroform solvent in the composite membrane was sufficiently volatilized and PLA/TiO$_2$ nanocomposite membranes of different mass concentrations were obtained.

## 2.4. Morphological Characterization

The morphologies of the nanofibers and films were observed using a high-resolution scanning electron microscope (SEM; Su 66000, Hitachi, Japan) operating at an accelerating voltage of 20 kV. The sample was cut into 2 × 2 mm$^2$ small pieces and fixed on a clean circular metal tray. Then a thin layer of gold was evenly sprayed onto the surface of the film under vacuum conditions. Finally, the morphology of the samples was observed under the SEM. The TiO$_2$ nanoparticles incorporated into PLA were subjected to transmission electron microscopy (TEM; JEM-2010 HR-TEM, JEOL, Tokyo, Japan) for morphological analysis.

## 2.5. Film Density Measurement

A thickness meter was used to measure the thicknesses of five regions of the sample film: the four corners and the center on the sample film. The weight m was measure using an analytical balance, and the measured length and width were *l* and *d*, respectively. The density (ρ) was measured three times were parallel measured by the samples in each group and calculated by the following formula:

$$\rho = m/l \cdot d \cdot a \tag{1}$$

## 2.6. FTIR Analysis

Fourier transform infrared (FTIR) analysis was performed to investigate the interactions between the main components of the film-forming materials. The film samples were characterized by FTIR spectroscopy (FTIR-650, Suzhou Leiden Scientific Instrument Co., Ltd., Suzhou, China) at 23 °C and 62.3% RH. The spectra in the range of 4000–650 cm$^{-1}$ with automatic signal gain was collected in 32 scans at 4 cm$^{-1}$ spectral resolution. To fully illustrate the result, each film sample was measured at three stochastic locations.

## 2.7. Tensile Properties

The tensile strength (TS) and elongation at break (EB; in percentage) of the nanofibers were measured using a texture analyzer (TA x$^T$ plus 50, Stable Micro Systems Ltd., Vienna, UK). Film staples (60 mm × 10 mm) were cut into a uniform size and followed the settings to test the tensile properties: pre-test speed of 20 mm/s, test speed of 3 mm/s, post-test speed of 10 mm/s, distance of 50 mm, and trigger force of 10 g, with the probe attached to a 5 kg load cell. Before the test, the samples were placed in a 50% RH chamber at 23 °C for 48 h.

### 2.8. Measurement of Water Solubility

The water solubility of the sample films was measured by employing the method recommended in ref. [11]. The film samples were cut into 2 cm × 2 cm rectangles, dried at 105 °C for 24 h, and the initial dry weight ($W_i$) was measured. Then put them into 50 mL of distilled water at 25 °C and agitated for 24 h, filter the solutions through a Whatman No. 1 filter paper. Next, dry the papers in a forced-air oven (105 °C, 24 h) and measured the final dry weight ($W_f$). The film solubility (%) was calculated by the following formula:

$$\text{Solubility (\%)} = (W_i - W_f) \times 100/W_i \tag{2}$$

### 2.9. WVP Measurement

The WVP of the bilayer films was gravimetrically determined according to previous reported [12]. Cut the films into a certain size and seal them on the top of a glass permeation cell (internal diameter = 3 cm). Use the silica gel to preserve the desiccator at 20 °C and 1.5% RH (28.044 Pa water vapor pressure) which made the cell involved distilled water (100% RH; 2.337 × 103 Pa vapor pressure at 20 °C) and stir the air in the desiccators. To determine the permeated weight loss of cell, the sample films were mensurated at intervals of 48 h for 10 days. This part of the water adsorbed by the desiccant was regarded as water that permeates out of the film, and could also be considered as a loss of cell weight. The slope of the functional relationship between the weight loss and the time curve is obtained by a linear regression equation. The WVP was calculated by the following formula:

$$\text{WVP} = (C \cdot X / A \cdot \Delta P) \tag{3}$$

where $C$ is the transmission rate of water vapor (gls), $X$ is the mean film thickness (mm), $A$ is the area of exposed film (mm$^2$), and $\Delta P$ is the partial water vapor pressure difference (Pa) between the both sides of the film. Repeat three tests for each sample to determine repeatability.

### 2.10. Contact Angle Measurement

A sessile drop assay was performed to measure the water contact angle on the samples by optical contact angle measurement. A water droplet (~20 μL) was gently deposited onto the surface of the nanofiber films, and photographs of the film surface were taken 5 s after droplet deposition. The angle between the water droplet and the surface of the film was confirmed by Image J software (version Java 1.6.0_05).

### 2.11. Light Transmittance and Opacity Measurements

The UV-vis transmission spectra of the film specimens were recorded in the range of 200~800 nm using a UV-vis spectrophotometer (UV-1800 (PC), Shanghai AuCy Instrument Co. Ltd., Shanghai, China). The opacity of the film specimens with well-controlled thicknesses that were put in the spectrophotometer test cell was calculated by the following formula [13]:

$$\text{Opacity} = \text{Abs600}/X \tag{4}$$

where Abs600 is the absorbance at 600 nm, and $X$ is the film thickness (mm). Convert the absorbance values to transmittance values using the Lambert-Beer equation. The measurement was repeated at least three times.

### 2.12. Antimicrobial Test

The antibacterial performance of the composite membrane against *Escherichia coli* (*E. coli*) and *Staphylococcus aureus (S. aureus)* were investigated by the inhibition zone method [14]. The film specimens were cut into small pieces of 10 mm diameter and irradiated for 48 h under UV-A light before fully activating the photocatalytic properties of TiO$_2$. The light source of UV radialization was

an 8 W Philip Cleo fluorescent lamp (Philips Co., Amsterdam, Holland) operating at its maximum intensity at 360 nm (i.e., the UV-A region) for 6 h. The vertical distance of illumination was 10 cm. Moreover, to verify the effects of the light source on the antibacterial ability of $TiO_2$, the experiment was respectively carried out under ultraviolet light irradiation and non-irradiation conditions. The UV irradiation of PLA/$TiO_2$ nanofibers was actualized in a CTC 256 climate chamber (Memmert (Shanghai) Trading Co., Ltd., Shanghai, China) maintained at 25 °C and a relative humidity of 70%. In order to have a certain amount of water molecules on the surface of $TiO_2$, the environment needed to maintain a high relative humidity. The bacteria were placed in a solid incubator at 37 °C and inoculated for 24 h to allow the bacteria to enter a stable log phase. After the test bacteria were activated, three gradients were diluted with physiological saline, and then 100 μL of microbial cell suspension having a concentration of $10^{-3}$–$10^{-4}$ ($10^5$–$10^6$ CFU/mL) was uniformly spread on the surface of the solid medium with an applicator until the surface dried. Then, a disc membrane was placed in the middle of the substrate. Finally, the medium was cultured in a constant-temperature incubator for 48 h at 37 °C. The bacteriostatic effect was estimated by measuring the diameter of the inhibition zone. The measurement was repeated at least three times.

## 3. Results and Discussions

### 3.1. Morphological Characteristics of PLA/TiO₂ Nanofibers and PLA/TiO₂ Films

Figure 1 shows that the nanofibers have a dimeter of 500–900 nm, which increased with the accretion of nano-$TiO_2$ particles. Figure 2 shows the morphology of the PLA/$TiO_2$ films. As can be seen, the surface roughness increased with an increase in $TiO_2$ nanoparticle content. Besides, with an increase in the concentration of $TiO_2$ nanoparticles, their dispersibility decrease, thereby inducing agglomeration of nanoparticles. PLA/$TiO_2$ nanofibers exhibited a good appearance when combined with nano-$TiO_2$ at a concentration of 1.25 wt.% or less. However, when the concentration reached 1.50 wt.%, the characteristic fiber morphology was significantly changed owning to the loading of nano-$TiO_2$. For the PLA/$TiO_2$ nanocomposite film, the smooth surface characteristic feature of the film changed when nano-$TiO_2$ with a concentration of 1.0 or higher was loaded. The same phenomenon was observed in the experiment carried out by Xie et al. [9]. They found that incorporated nano-$TiO_2$ with PCL could form the porous structure and accumulation of $TiO_2$ nanoparticles which result in the uneven surface. Nevertheless, the PLA matrix acted as a good stabilizing agent for the $TiO_2$ nanoparticles, as indicated by the spherical discrete particles in the SEM images of both PLA/$TiO_2$ nanofibers and films.

The TEM images show the $TiO_2$ nanoparticles and their uniform distribution on the PLA fibers and films. The $TiO_2$ nanoparticles are spherical with a uniform morphology. Among the all composite fibers and films with different concentrations of $TiO_2$, the PLA/$TiO_2$-0.75 nanofibers and films showed the best dispersion of $TiO_2$ nanoparticles (Figure 3). The nano-$TiO_2$ nanoparticles which were more uniformly dispersed in the nanofibers and the nanocomposite films exhibited a good mosaicism, and did not affect the normal morphology of the nanofibers and the nanocomposite membranes while functioning. In addition, the particle size, estimated from the selected area electron diffraction pattern, was found to be 10–20 nm, which was verified by light transmittance and opacity data. The similar result was reported by Toniatto et al. who discovered that through the electrospinning progress, a homogeneous distribution of nano-$TiO_2$ enabled with PLA was formed [8].

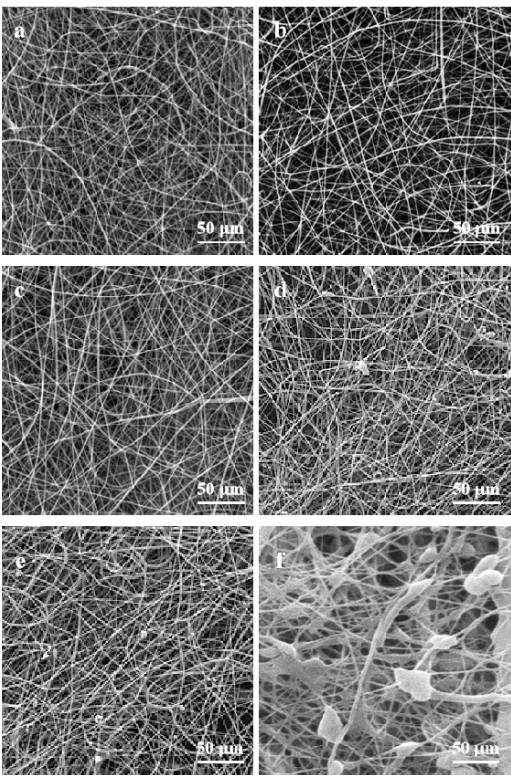

**Figure 1.** SEM images of polylactic acid (PLA)/TiO$_2$ nanofibers: (**a**) pure PLA; (**b**) PLA/TiO$_2$-0.5; (**c**) PLA/TiO$_2$-0.75; (**d**) PLA/TiO$_2$-1.0; (**e**) PLA/TiO$_2$-1.25, and (**f**) PLA/TiO$_2$-1.5.

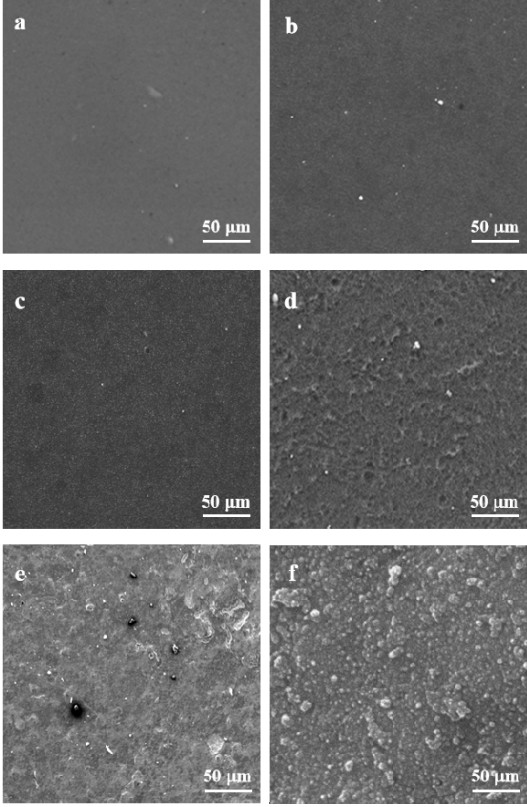

**Figure 2.** SEM images of PLA/TiO$_2$ films: (**a**) pure PLA; (**b**) PLA/TiO$_2$-0.5; (**c**) PLA/TiO$_2$-0.75; (**d**) PLA/TiO$_2$-1.0; (**e**) PLA/TiO$_2$-1.25, and (**f**) PLA/TiO$_2$-1.5.

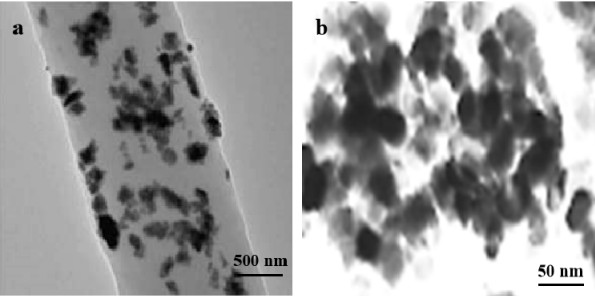

**Figure 3.** TEM images of (**a**) PLA/TiO$_2$-0.75 nanofiber and (**b**) PLA/TiO$_2$-0.75 film.

### 3.2. Film Density

The density of the nanofibers and films with different TiO$_2$ contents are displayed in Table 1. Both the PLA/TiO$_2$ nanofibers and films were thicker than the control film; however, there was no significant difference in their thicknesses ($p > 0.05$). Due to the addition of nanosized TiO$_2$ particles into the PLA matrix with a dense chemical structure, the internal space network structure of the PLA polymer was destroyed, causing an overall density reduction [15]. A similar result was presented for whey protein isolate films containing porous silica-coated nano-TiO$_2$ particles as a reinforcing medium [16]. Nevertheless, the density of the PLA/TiO$_2$-1.25 nanofibers and films increased slightly; this is possibly because of the increase in the amount of added TiO$_2$, which is difficult to disperse and likely to agglomerate in PLA, resulting in a slightly increased density than that of the former. A similar result was obtained by the addition of TiO$_2$ nanoparticles and high-pressure hydrodynamics to improve the functions of a PVA/CS composite film [17].

**Table 1.** Density of PLA/TiO$_2$ nanofibers and films with different contents of TiO$_2$.

| Sample/Density (g/cm$^3$) | Nanofiber | Film |
|---|---|---|
| PURE PLA | $0.98 \pm 0.03$ [a] | $1.01 \pm 0.04$ [b] |
| PLA/TiO$_2$-0.5 | $0.64 \pm 0.08$ [b] | $0.73 \pm 0.03$ [a] |
| PLA/TiO$_2$-0.75 | $0.30 \pm 0.07$ [c] | $0.33 \pm 0.01$ [b] |
| PLA/TiO$_2$-1 | $0.28 \pm 0.04$ [d] | $0.27 \pm 0.01$ [c] |
| PLA/TiO$_2$-1.25 | $0.36 \pm 0.03$ [c] | $0.43 \pm 0.02$ [a] |
| PLA/TiO$_2$-1.5 | $0.39 \pm 0.09$ [b] | $0.55 \pm 0.03$ [d] |

a, b, c, d: Means with different letters within a column indicate significant differences ($p \leq 0.05$).

Furthermore, the density of the nanofibers was lower than that of the films, which could be ascribed to the small average size of TiO$_2$ nanoparticles distributed on the PLA nanofibers. Even though tens of thousands of fibers were interwoven to form the nanofiber, the resulting network was porous due to the pores between the fibers, which reduced the density as compared with that of the films. On the contrary, the film was cast from a mixture of PLA solution and TiO$_2$ dispersion liquid. The intermolecular crosslinking was strong, and the pores were occupied by PLA molecules; as a result, the films were denser than the nanofibers.

### 3.3. Contact Angle

The water contact angles of the PLA/TiO$_2$ nanofibers and films are shown in Figure 4. As can be seen, both PLA/TiO$_2$ nanofibers and films show the same trend of change in water contact angle. The water contact angle of 138.4° ± 5.2° of the PLA fibers is due to the hydrophobicity of the PLA fibers, which is evident from the appearance of spherical water droplets on the fiber surface for a long period. The water contact angle of the PLA film (113.5° ± 2.7°) was slightly lower than that of the PLA fibers. This is probably because the film surface was smoother than the nanofiber surface, which led to its greater interaction with water, thereby resulting in a smaller water contact

angle. The water contact angles of the nanofiber and film slightly decreased due to the addition of 0.75 wt.% $TiO_2$ (122.1° ± 6.2° and 98.7° ± 4.4°, respectively). The water contact angle decreases with incremental surface hydrophilicity. Considering this, the reduced contact angle measured in this study may be attributed to the lower hydrophilicity of $TiO_2$, which might result in the augment of surface hydrophilicity. Zhang et al. also reported that the contact angle decreases because of the hydrophilic nature of the incorporated $TiO_2$ nanoparticles, which resulted in significantly enhanced hydrophilicity [18]. Alternatively, the decrease in contact angle upon addition of $TiO_2$ nanoparticles can be attributed to the increases in surface free energy and surface roughness of the film. However, the contact angle of the nanofibers increased to 143.4° ± 4°, while that of the films slightly increased to 119.1° ± 5° with a further increase in the $TiO_2$ content to 1.25 wt.%. However, beyond 1.25 wt.% of $TiO_2$, no significant difference was observed in the water contact angles. This is because increasing the amount of $TiO_2$ nanoparticles results in particle agglomeration in the film, which increases the hydrophobicity of the film. Salarbashi et al. [19] claimed that combined the $TiO_2$ with soluble soybean polysaccharide could increase hydrophobicity of bionanocomposite film which also authenticated our results. Similar findings were reported by Alippilakkotte et al. [20]. Moreover, some papers reported that owing to the increased surface hydrophobicity, accreting the amount of $TiO_2$ nanoparticles tends to prevent the water drop from spreading over the film surface [21,22].

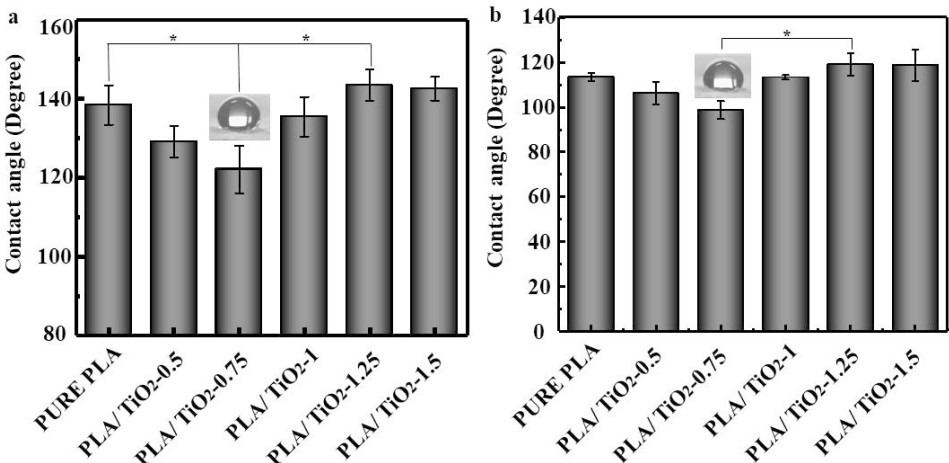

**Figure 4.** Variation in water contact angle with the addition of $TiO_2$ nanoparticles in (**a**) PLA nanofibers and (**b**) PLA films. Columns with * mean significantly different ($p \leq 0.05$).

*3.4. FTIR Study*

The molecular interaction between the raw materials and $TiO_2$ nanoparticles was studied by Fourier transform infrared spectroscopy (FTIR). As there was no difference in the functional groups of the nanofiber and film, as observed by FTIR analyses, a pure PLA fiber and a PLA/$TiO_2$ film were used as representatives for drawing. The FTIR spectra of $TiO_2$ nanoparticles, pure PLA fiber, and PLA/$TiO_2$ film are shown in Figure 5. The $TiO_2$ spectrum shows weak peaks around 1630 cm$^{-1}$ and 3440 cm$^{-1}$, which arise from the stretching vibrations of surface O–H groups of $TiO_2$ [23]. In addition, the vibration band at 1110 cm$^{-1}$ originates from the stretching of C–O–C bonds [24]. In the pure PLA fiber spectrum, the weaker peak at 2925 cm$^{-1}$ appears due to the asymmetric stretching vibrations of –CH$_3$ and –CH$_2$, the absorption peak at 1750 cm$^{-1}$ arises from the stretching vibration of aldehyde or ester carbonyl C=O [25], and the peak at 1080 cm$^{-1}$ arises from the stretching vibrations of C–O–C bonds and C–O bonds [26]. In the spectrum of the PLA/$TiO_2$ film, the band at around 3420 cm$^{-1}$ is caused by the tensile vibration of the characteristic hydroxyl group of $TiO_2$ nanoparticles, and shows a slight shift from 3440 cm$^{-1}$. This is possibly because the mixing of PLA with $TiO_2$ led to partial incorporation of a large amount of hydroxyl groups present on the $TiO_2$ surface into the composite membrane, resulting in the appearance of the vibration peak of hydroxyl group in the composite membrane spectrum. In addition,

the peak at 2925 cm$^{-1}$ arising from the vibrations of –CH$_3$ and –CH$_2$ of pure PLA film shifted to 2930 cm$^{-1}$. The addition of nanoparticles kept the signal at 1750 and 1080 cm$^{-1}$ unchanged, which might probably due to the physical mixing of PLA and TiO$_2$. However, the intensity of absorption peak and the area under the peak increased for the PLA/TiO$_2$ specimens compared with the pure PLA fiber. Li et al. observed a similar increase in peak intensities [27]. Overall, no new peaks were detected for the PLA/TiO$_2$ nanocomposite films, which indicated that except for the surface hydroxyl group present on the TiO$_2$ nanoparticles, the components were only physically mixed. Besides, no functional group changes occurred, and the base structure of the films was not destroyed.

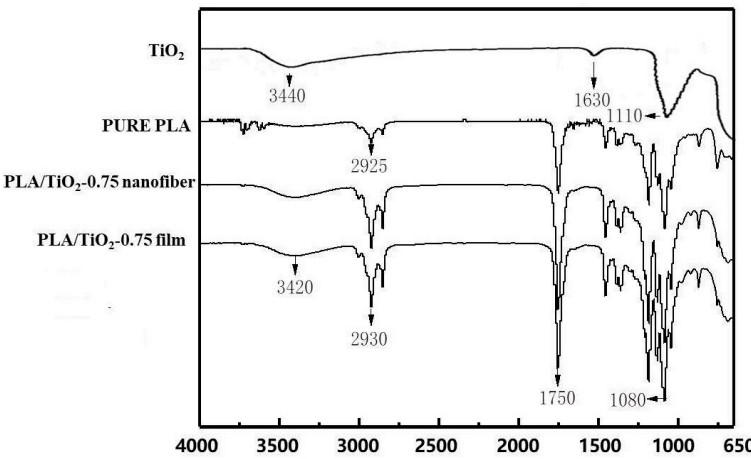

**Figure 5.** FTIR spectra of TiO$_2$ nanoparticles, pure PLA, PLA/TiO$_2$-0.75 nanofiber, and PLA/TiO$_2$-0.75 film.

*3.5. Water Solubility*

As shown in Table 2, the solubility of the nanofibers and nanocomposite films significantly increased as the TiO$_2$ content increased from 0 to 1.0 wt.%. Owning to the hydrophilicity of nano-TiO$_2$, the water solubility of the nanocomposite was higher than that of the hydrophobic pure PLA film. However, as the TiO$_2$ content of the nanocomposite films was increased to 1.25 and 1.50 wt.%, the solubility of the films significantly decreased ($p < 0.05$). This trend can be accounted for the higher concentration of TiO$_2$ nanoparticles (1 wt.%), which reduced the space available to host water molecules, and/or result in the stronger interactions between nano-TiO$_2$ particles and PLA that led to the TiO$_2$ particles acting as a crosslinker, thereby arresting water access [28,29]. A parallel observation was reported by Alizadeh-Sania, who demonstrated that high levels of film swelling by water inhibit their biodegradability. Besides, it could be observed that the solubility of nanofibers was higher than that of the nanocomposite films, which is probably because electrospinning led to uniform spinning of the nanofibers, which could fully distribute the nanosized TiO$_2$ [30]. On the contrary, during the solution casting process, TiO$_2$ nanoparticles were more likely to aggregate, which reduced the specific surface area of TiO$_2$ as compared with that of the nanofibers; hence, its hydrophilicity could not give full play to the effect [31].

**Table 2.** Density of PLA/TiO$_2$ nanofibers and films with different contents of TiO$_2$.

| Sample/Water Solubility (%) | Nanofiber | Film |
|---|---|---|
| PURE PLA | 1.12 ± 0.3 [a] | 1.08 ± 0.2 [b] |
| PLA/TiO$_2$-0.5 | 9.57 ± 0.8 [b] | 4.82 ± 0.4 [a] |
| PLA/TiO$_2$-0.75 | 15.63 ± 0.7 [c] | 9.46 ± 0.6 [b] |
| PLA/TiO$_2$-1 | 34.09 ± 1.4 [d] | 19.29 ± 0.9 [c] |
| PLA/TiO$_2$-1.25 | 16.87 ± 1.3 [c] | 10.14 ± 0.3 [a] |
| PLA/TiO$_2$-1.5 | 8.80 ± 0.9 [b] | 5.05 ± 0.2 [d] |

a, b, c, d: Means with different letters within a column indicate significant differences ($p \leq 0.05$).

### 3.6. Water Vapor Permeability

Figure 6 shows the effects of nanoparticle content on the WVP of the nanofibers and films. The WVP of the PLA/TiO$_2$ nanofibers and films exhibited the same trend and decreased significantly with an enhancive content of TiO$_2$ from 0 to 1 wt.%. However, further increasing the TiO$_2$ content increased the WVP of the nanofibers and films. This is because the hydrophobic TiO$_2$ nanoparticles prevented the micro- and nano-paths in the network structure [32]. At low contents, the nano-TiO$_2$ particles could properly separate throughout the matrix, thereby blocking the entry of water vapor. The reason why WVP appears to drop significantly after adding nanoparticles could be primarily attributed to the highly complex path that was generated because of the distribution of impermeable TiO$_2$ nanoparticles throughout the matrix. Generally, the nanoparticles increase the pass length of water molecules by compelling the permeating molecules to randomly oscillate around them [33]. As previously mentioned, the number of available OH groups reduced because of the production of hydrogen bonds between Ti–O and the hydroxyl groups of PLA; this consequently reduced the WVP of the films. Moreover, water-insoluble TiO$_2$ tends to hinder the ability of water vapor molecules in the membrane, thereby prolonging the path of water vapor movement and reducing water vapor permeation. However, high TiO$_2$ contents (1.25 and 1.5 wt.%) resulted in nanoparticle aggregation, destroying the structure of the film [34]. Kadam et al. [16]. observed a similar result, while El-Wakil et al. [35]. demonstrated that increasing the TiO$_2$ nanoparticle content to 1 wt.% decreased the WVP of the film by 10%. They also reported that the high surface energy caused by excess TiO$_2$ caused the nanoparticles to agglomerate (as shown by SEM) at higher nanoparticle concentrations; this consequently reduced the effectual concentration of the nanoparticles and promoted WVP [16]. Baek et al. reported that the WVP was lower when the nanoparticles were more stably dispersed throughout the PLA films [36].

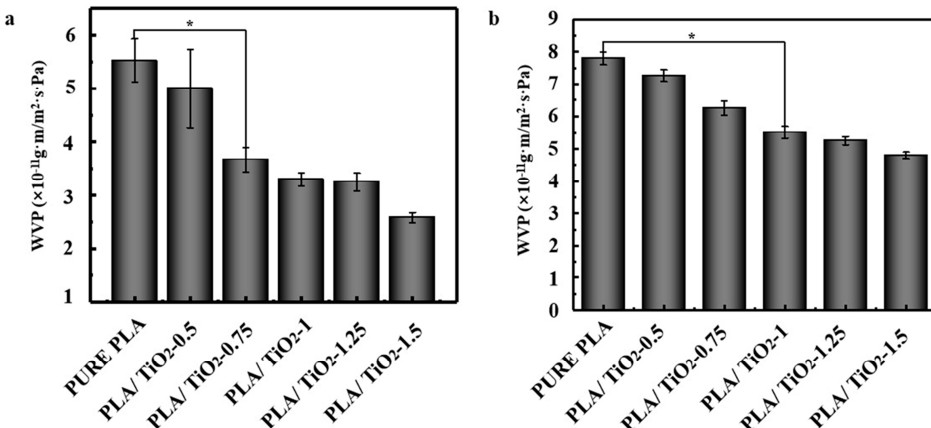

**Figure 6.** WVP of PLA/TiO$_2$ nanofibers and films with different TiO$_2$ contents: (**a**) nanofibers; (**b**) films. Columns with * mean significantly different ($p \leq 0.05$).

It was found that the water vapor transmission rates of the films were much larger than those of the nanofibers. Due to the extremely small diameter of the nanofibers, the fine structure of the fiber could be effectively regulated. With PLA as the film-forming substrate, the prepared nanofiber had superior hydrophobic properties. In contrast, the casting film was prepared by simple physical mixing and ultrasonic dispersion. TiO$_2$ was embedded in the interstitial spaces of PLA, which increased the crystal gap of PLA molecules, and water molecules could more easily pass through the composite membrane, thus reducing the hydrophobicity of PLA and increasing the water vapor transmission rate of the film.

### 3.7. Light Transmittance and Opacity

Tables 3 and 4 show the variation in the transmittance (*T*%) of pure PLA and PLA/TiO$_2$ nanofibers and films in the UV-C (240 nm), UV-B (300 nm), and UV-A (360 nm) wavelength ranges. As can

be seen, with an increase in $TiO_2$ content, the transmittance of the nanofibers in the UV-A, UV-B, and UV-C regions first increased and then decreased, which resulted in the agglomeration of $TiO_2$ nanoparticles, and reached maximum values of 29.64% ± 0.91%, 28.45% ± 1.92%, and 28.16% ± 3.08%, respectively, for PLA/$TiO_2$-1.0. The nanocomposite film exhibited the same trend in the UV-B and UV-C regions; however, the highest transmittance values were 0.81% ± 0.06% and 1.90% ± 0.01% for the PLA/$TiO_2$-0.75 film. The transmittance of the nanofibers was higher than that of the films because the nanofibers were porous and less dense than the films. Further, the $TiO_2$ content affected the amount of UV-A light that can transmit through the film. The accretion of $TiO_2$ to PLA fibers and films reduced all the UV transmissions significantly ($p < 0.05$), thus showing strong UV light-blocking effects because $TiO_2$ is a kind of whitening agent and a semiconductor that has an optical band gap of 385 nm, and scatters visible light [14].

**Table 3.** UV light opacity of PLA-based nanofibers as a property of $TiO_2$ content.

| Fiber | UVC (240 nm) | UVB (300 nm) | UVA (360 nm) |
|---|---|---|---|
| PURE PLA | 24.91 ± 1.90 [a] | 19.83 ± 2.00 [c] | 19.46 ± 1.22 [c] |
| PLA/$TiO_2$-0.5 | 21.63 ± 2.28 [a] | 19.26 ± 1.90 [c] | 19.07 ± 1.29 [c] |
| PLA/$TiO_2$-0.75 | 23.66 ± 1.83 [a] | 22.01 ± 2.55 [a] | 22.09 ± 2.90 [a] |
| PLA/$TiO_2$-1 | 29.64 ± 0.91 [b] | 28.45 ± 1.92 [b] | 28.16 ± 3.08 [b] |
| PLA/$TiO_2$-1.25 | 17.17 ± 1.63 [c] | 16.16 ± 1.76 [d] | 16.12 ± 2.30 [d] |
| PLA/$TiO_2$-1.5 | 16.69 ± 0.58 [c] | 15.91 ± 1.92 [c] | 14.58 ± 2.57 [c] |

a, b, c, d: Means with different letters within a column indicate significant differences ($p \leq 0.05$).

**Table 4.** UV light opacity of PLA-based films as a function of $TiO_2$ content.

| Film | UVC (240 nm) | UVB (300 nm) | UVA (360 nm) |
|---|---|---|---|
| PURE PLA | 32.4 ± 2.15 [a] | 46.86 ± 0.60 [a] | 59.92 ± 0.70 [a] |
| PLA/$TiO_2$-0.5 | 0.30 ± 0.05 [b] | 0.44 ± 0.06 [c] | 66.31 ± 11.2 [a] |
| PLA/$TiO_2$-0.75 | 0.81 ± 0.06 [b] | 1.90 ± 0.01 [b] | 61.27 ± 4.10 [a] |
| PLA/$TiO_2$-1 | 0.25 ± 0.01 [b] | 0.25 ± 0.65 [c] | 69.88 ± 1.65 [a] |
| PLA/$TiO_2$-1.25 | 0.20 ± 0.01 [b] | 0.19 ± 0.05 [c] | 62.34 ± 3.05 [a] |
| PLA/$TiO_2$-1.5 | 0.18 ± 0.04 [c] | 0.17 ± 0.03 [c] | 60.78 ± 2.31 [c] |

a, b, c: Means with different letters within a column indicate significant differences ($p \leq 0.05$).

Xie et al. fabricated PLA/$TiO_2$ nanoparticle biodegradable films via a solution casting method. They reported that the UV-A transmission of the PLA film decreased from 87.67% to 0.68% when the $TiO_2$ content was increased from 0% to 5% [9]. The transmittance of the prepared PLA/$TiO_2$ film was lower than that of the pure PLA film in the visible region, which affects light absorption and scattering. Similar results have been reported on the difference of the UV light transmittance of chitosan and gelatin/agar films after combined with $TiO_2$ nanoparticles [32].

Table 5 shows the opacity of the nanofibers and films as a property of $TiO_2$ nanoparticle content. The results showed that the metallic nature of the $TiO_2$ nanoparticles allows both nanofibers and films to effectively act as a barrier to the penetration of UV light while increasing their opacity. Since $TiO_2$ nanoparticles were present as a whitening agent in the polymer matrix, it aggrandized the opacity, increased whiteness, and reduced the transparency of the film. However, in the case of nanofiber, the mildly augment in opacity observed at low nano-$TiO_2$ contents may be resulted in the absorptive or reflective functions of the nanoparticles. Additionally, the increase in opacity at high $TiO_2$ contents is attributable to the partial agglomeration and self-assembly of the nano-$TiO_2$ particles in the matrix, which hinder light from passing through the bilayer membranes [31]. In the case of films, PLA/$TiO_2$-0.75 exhibited the lowest opacity among all the films with different $TiO_2$ contents. The opacity of the other films was higher than that of the pure PLA film; this is possibly because of the dispersing of nano-$TiO_2$ in the PLA membranes, possessed larger surface of nano-$TiO_2$ for UV light blocking which blocked UV rays and had a better shielding effect on visible light. According

to Mallakpour et al., PVA combined with modified nano-$TiO_2$ exhibited remarkably high absorption and increased the opacity of the composite membranes, which was around the UV region indicating the result. They considered that this type of nanocomposite film could be used as a coating to shield against UV light [26]. He et al. [37] fabricated the fish skin gelatin/$TiO_2$ nanocomposite films. They observed that compound with nano-$TiO_2$ could decrease the light transmittance of gelatin, however, could effectively prevented the UV light which demonstrated our results.

**Table 5.** Visible light transmittance percentage (%) and opacity of PLA-based nanofibers and films as a function of $TiO_2$ content: (a) nanofibers; (b) films.

| Membrane | (a) Nanofibers | | (b) Films | |
|---|---|---|---|---|
| | Visible (600 nm) (%) | Opacity | Visible (600 nm) (%) | Opacity |
| PURE PLA | 0.96±0.02 [a] | 18.83 ± 2.01 [a] | 66.5 ± 2.20 [a] | 3.10 ± 0.35 [a] |
| PLA/$TiO_2$-0.5 | 0.48 ± 0.05 [b] | 19.93 ± 0.94 [a] | 0.67 ± 0.71 [b] | 15.57 ± 0.36 [a] |
| PLA/$TiO_2$-0.75 | 0.46 ± 0.03 [b] | 22.26 ± 0.90 [b] | 3.05 ± 0.01 [b] | 8.94 ± 0.059 [b] |
| PLA/$TiO_2$-1 | 0.46 ± 0.01 [b] | 30.09 ± 2.01 [c] | 0.54 ± 0.06 [b] | 17.46 ± 0.38 [c] |
| PLA/$TiO_2$-1.25 | 0.44 ± 0.02 [b] | 17.59 ± 1.00 [a] | 0.44 ± 0.05 [b] | 14.13 ± 0.29 [a] |
| PLA/$TiO_2$-1.5 | 0.43 ± 0.01 [b] | 17.68 ± 0.43 [a] | 0.32 ± 0.02 [b] | 12.38 ± 0.20 [a] |

a, b, c: Means with different letters within a column indicate significant differences ($p \leq 0.05$).

### 3.8. Mechanical Properties

Figure 7 illustrates the mechanical properties (TS and EB) of PLA/$TiO_2$ nanofibers and films with different $TiO_2$ contents. As can be seen, the TS of the 0.5 wt.% $TiO_2$ nanofiber decreased by 2.31 ± 0.12 to 1.72 ± 0.11 MPa. Athanasouliaa et al. found that incorporating $TiO_2$ into a PLA matrix decreases the TS and EB of the nanocomposites [38]. Upon increasing the $TiO_2$ nanoparticle content to 0.75%, the TS of the nanofibers increased from 1.72 ± 0.11 to 2.71 ± 0.12 MPa. The TS enhanced resulting in an augment in the internal friction in the matrix because of the addition of $TiO_2$ nanoparticles, which created a crystalline structure [39]. Because of its van der Waals interaction with the hydroxyl groups of PLA molecules, the concentration of filler particles, PLA crystallinity, and the interfacial adhesion of the fillers must be carefully considered to enhance the strength [18]. The substantial TS reduction that was observed at high $TiO_2$ concentrations may be attributed to the self-networking of nanoparticles that led to their agglomeration and non-homogeneous dispersion, as this phenomenon could reduce the longitudinal and lateral TS of the nanofibers [31]. Besides, Habiba et al. argued that the addition of $TiO_2$ nanoparticles into chitosan/PVA nanofibrous membranes could increase the toughness of the nanofibrous membranes owning to the stable polymer–nanoparticle interface that enhances the mechanical property [40].

The EB is impacted by the volume fraction and distribution of the fillers in the matrix, meanwhile, their interaction with each other could also affect [12]. Generally, the EB of nanocomposites decreases at a high filler content. In our study, the EB of the PLA/$TiO_2$ nanofibers in the lateral and longitudinal directions gradually decreased from 57.5% ± 2.5% to 17.5% ± 2.1%. This phenomenon is ascribed to the anti-plasticization. Nano-$TiO_2$ particles might play the part of an anti-plasticizer due to increased interaction, lessening the free volume between the biopolymer chains and reducing film flexibility. Similar results have been reported by Fonseca et al. and Shaili et al. [41]. Thus, as reinforcement and agglomeration were respectively observed at lower and higher $TiO_2$ concentrations, 0.75 wt.% was considered the optimal $TiO_2$ nanoparticle filler concentration for a PLA fiber matrix.

In the case of films, with an increase in $TiO_2$ content from 0% to 0.75%, the TS of the PLA/$TiO_2$ films significantly increased from 3.16 ± 0.15 MPa to 14.49 ± 0.47 MPa ($p < 0.05$). A similar result has been reported in the latest research [42]. In contrast, the EB decreased from 62.71% to 40.61% with an augment in $TiO_2$ content from 0% to 0.5%. At a $TiO_2$ content of 0.75%, the EB increased to 49.03%, and then decreased sharply to 33.23% and 27.05%, significantly ($p < 0.05$). This result was consistent with that reported by He et al., who reported that the TS of a gelatin casting film containing a certain

amount of TiO$_2$ nanoparticles increased owing to strengthening of the association between protein molecule and metal oxide nanoparticles [37].

The reciprocities between PLA and TiO$_2$ enhanced as the quantity of TiO$_2$; therefore, it caused the agglomeration and inhomogeneous dispersion of nanoparticles. At higher TiO$_2$ contents of over 0.75 wt.%, the agglomeration of the nanoparticles was intensified, thereby destroying the mechanical equilibrium system, hence reducing the TS and EB of the film. Olevaei et al. found that both the TS and EB of a starch composite containing TiO$_2$ nanoparticles and montmorillonite at high concentrations decreased [12]. It was observed that the TS and EB values of the PLA/TiO$_2$ membranes were higher than those of the pure PLA film. This indicates that the dispersion of TiO$_2$ in the film matrix has a synergistic effect, which enhances the microstructure of the film, thereby improving the TS of the PLA/TiO$_2$ film [34]. These results indicated the tiny size of TiO$_2$ nanoparticles made an important impact as a filler, filling the network structure of the macromolecules and promoting the formation of three-dimensional networks between the two materials.

The results indicated that the TS and EB of the nanofibers were lower than those of the films. Particularly, the TS of the films were about 367% higher than that of the nanofibers because the nanofibers were mainly subjected to local stress during tensile testing that resulted in a high load torque on each fiber. However, the film was subjected to overall forces, and the tight crosslinking between the molecules rendered its mechanical properties more stable. Therefore, it could be concluded that the mechanical properties of the film are better than those of the nanofiber.

According to these results, 0.75 wt.% TiO$_2$ content of the PLA film matrix is sufficiently to acquire well-balanced mechanical properties in both elastic and plastic regions, which is suitable for food packaging applications of both nanofibers and films.

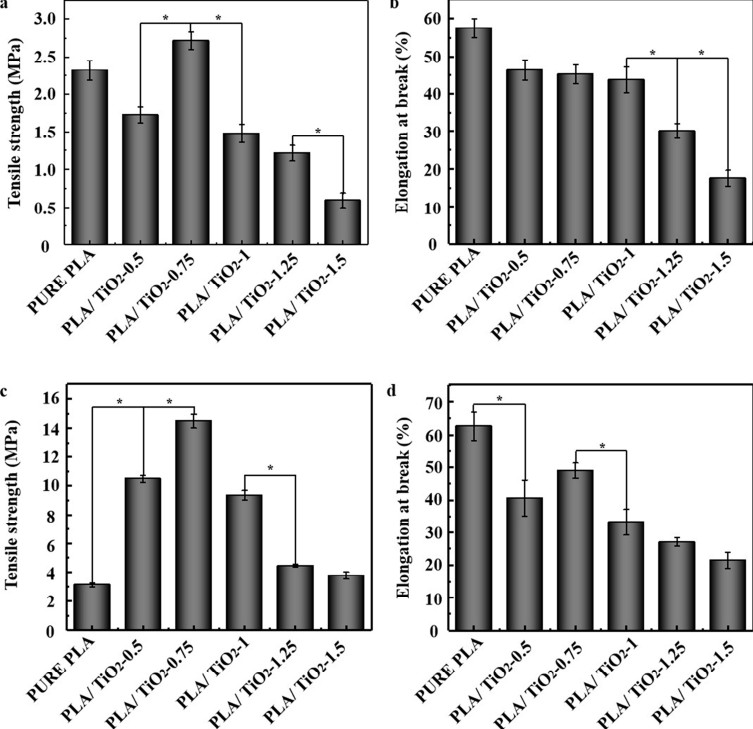

**Figure 7.** Variation in tensile strength (TS) and elongation at break (EB) of PLA nanofibers and films: (**a**,**b**) nanofibers; (**c**,**d**) films. Columns with * mean significantly different ($p \leq 0.05$).

### 3.9. Antibacterial Property

The antibacterial properties of the nanofibers and films were evaluated for two types of model bacteria, i.e., gram-negative (*E. coli*) and gram-positive (*S. aureus*) bacteria, under two different lighting conditions: a fluorescent lamp source and UV-A (360 nm) irradiation. As can be seen in Figure 8,

the pure PLA film has almost no antibacterial properties, whereas the antibacterial effect of the nanocomposite film increased gradually with increasing amounts of TiO$_2$ nanoparticles, and the highest inhibition ratio was achieved at a TiO$_2$ content of 0.75 wt.% for both nanofibers and films. At a TiO$_2$ content of 0.75 wt.%, the inhibition zones of the nanofibers for *E. coli* and *S. aureus* were 4.86 ± 0.50 and 5.98 ± 0.77 mm and improved by 1115% and 1268%, respectively, as compared with those of the pure PLA fiber. The inhibition zones of the films were 3.69 ± 0.40 and 4.63 ± 0.45 mm for *E. coli* and *S. aureus*, respectively. It is well known that TiO$_2$ and various TiO$_2$-based photocatalysts can only become active under light irradiation. Toniatto et al. reported that the bactericidal effect of electrospinning PLA/TiO$_2$ nanofibers against *S. aureus* was strong after UV irradiation, which activated the TiO$_2$ nanoparticles, and that the effect was influenced by the time associated to TiO$_2$ concentration [8]. Fonseca et al. measured the antimicrobial effect of PLA/TiO$_2$ nanocomposite films including 8 wt.% nano-TiO$_2$ under UV irradiation and reported the killing effect on bacteria and fungi achieved 94.3% and 99.9%, respectively [5].

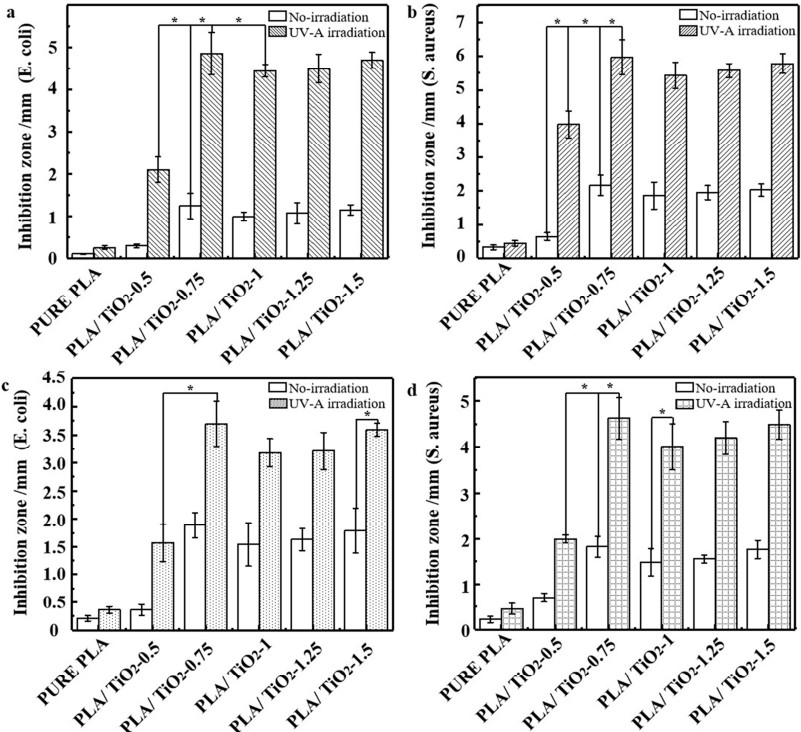

**Figure 8.** Inhibition zones of *E. coli* and *S. aureus* as a function of TiO$_2$ nanoparticle content in PLA nanofibers and films under light irradiation conditions: (**a**,**b**) nanofibers; (**c**,**d**) films. Columns with * mean significantly different ($p \leq 0.05$).

These consequences pointed that the antimicrobial activity of PLA/TiO$_2$ nanocomposites could be ascribed to the nano-TiO$_2$ particles; this is consistent with the findings of He et al. [39]. TiO$_2$ has strong oxidizing properties under long-wavelength ultraviolet light (320–400 nm), resulting in microbial death as shown by the experimental results. Particularly, active species such as hydrogen peroxide, hydroxyl radicals, and superoxide anions are generated, which might passivate the microorganisms through the process of cell lysis that affects the genome and other intracellular molecules when the irradiated TiO$_2$ nanoparticles contacts the microbes. Thus, higher TiO$_2$ contents result in greater generation of ROS and a substantially increased antibacterial rate. Li et al. demonstrated that the bactericidal effect of the composite films was affected by the lighting conditions in their study [43].

The results of this study show that nano-TiO$_2$ particles are efficacious in restraining both Gram-positive and Gram-negative bacteria. However, the inhibition zones formed by the PLA/TiO$_2$ nanocomposite fibers and films for *S. aureus* are larger than those for *E. coli*. Similar results were

reported by Salarbashi et al. [20], who observed that after the bacterial inhibition test, the vitality of *E. coli* was higher than that of *S. aureus*, which supports our conclusion. They concluded that this was because of the different compositions of the cell walls of Gram-negative and Gram-positive bacteria; the cell wall of gram-positive bacteria consisted of a peptidoglycan layer with a thick wall thickness that effectively blocked the penetration of $TiO_2$ into the cell membrane, whereas the cell wall of gram-negative bacteria consisted of a thin layer of peptidoglycan and an inner and outer membrane. The structure was complex, so that it could protect bacteria from many chemical agents [10].

Due to the high specific surface area and high porosity of the nanofiber, nano-$TiO_2$ particles were evenly separated on the surface of the fiber membrane. Besides, the photocatalytic reaction of $TiO_2$ was achieved by direct contact between the free radicals and test bacteria; therefore, $TiO_2$ was fully activated and photocatalysis occurred on its surface. $TiO_2$ nanoparticles easily aggregated in the PLA molecules owning to the physical stirring and mixing, which reduced the photocatalytic reaction. Therefore, the inhibition effect of the nanofibers was more efficient than that of the films.

## 4. Conclusions

In this study, electrospinning and solution casting were performed to successfully prepare PLA/$TiO_2$ nanofibers and films, respectively. The water contact angle initially decreased and then increased with increasing $TiO_2$ content. Moreover, $TiO_2$ enhanced the solubility of the film and its ability to act as an effective barrier. Furthermore, the antibacterial activity improved under UV-A irradiation, and nanofibers and films with 0.75 wt.% $TiO_2$ content exhibited inhibition zones of 4.86 ± 0.50 and 3.69 ± 0.40 mm for *E. coli* and 4.63 ± 0.45 and 5.98 ± 0.77 mm for *S. aureus*, respectively. The best overall functionality was observed for a $TiO_2$ content of 0.75 wt.% for the PLA/$TiO_2$ nanofibers and films. The performance of the nanofiber was better than that of the film. Future studies are required to test the effectiveness of the films as packaging material for real food.

**Author Contributions:** Data Curation, S.A.; Writing—Original Draft Preparation, S.F.; Writing—Review & Editing, F.Z.; Project Administration, Y.L.; Funding Acquisition, Y.L.

**Funding:** This research was supported by Sichuan Science and Technology Program (2018RZ0034), and Natural Science Fund of Education Department of Sichuan Province (16ZB0044 and 035Z1373).

**Conflicts of Interest:** The authors declare no conflict of interest.

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
