# Peer review of "Physico-Mechanical and Antibacterial Properties of PLA/TiO2 Composite Materials Synthesized via Electrospinning and Solution Casting Processes"

_coatings, doi:10.3390/coatings9080525_

Round 1

Reviewer 1 Report

The authors gave fully detailed   report on physico-chemical and antibacterial properties of polylactic acid-titanium oxide (PLA/TiO2) composite materials synthesized via two methods. They claimed the composite materials have potential application in antimicrobial packaging. They tried to support their claim with various characterization and analytical tools. I have some minor comments and suggestions which should be included in the final article format.

In the past 5 years there have been some related works published in various journals using PLA/TiO2) composite materials for biocidal activity. Some even have very close topic with the current work despite some differences in their content. In the current work I would suggest the authers to compare their results with already published works on PLA/TiO2 composite materials in addition to comparing related materials like soybean polysaccharide/TiO2 bionanocomposite (ref 20) and the work of Alizadeh-Sani et al (ref 31). I commend the authors to include some typical figures for the contact angles measured You should change the topic of section 3 to result and discussions rather than ‘results’ Section 3.6 I would write the topic in full rather than abreviations ‘’WVP’’ In figure 5 the you can treat the spectrum of TiO2   some correction to make it horizontal like the other spectrum. Check the first sentence of the new paragraph in the line 332-333.

Author Response

Reviewer #1: Comments:

In the current work I would suggest the authers to compare their results with already published works on PLA/TiO2 composite materials in addition to comparing related materials like soybean polysaccharide/TiO2 bionanocomposite (ref 20) and the work of Alizadeh-Sani et al (ref 31). 

Re: Added. The references are as follows

The same phenomenon was observed in the experiment carried out by Xie et.al. They found that incorporated nano-TiO2 with PCL could form the porous structure and accumulation of TiO2 nanoparticles which result in the uneven surface. (Lwt., 2018, 96, 307-314)

2.The similar result was reported by Toniatto et al. who discovered that through the electrospinning progress, a homogeneous distribution of nano-TiO2 enabled with PLA was formed. (Mater. Sci. Eng. C Mater. Biol. Appl., 2017, 71, 381-385)

Salarbashi et al. claimed that combined the TiO2 with soluble soybean polysaccharide could increase hydrophobicity of bionanocomposite film which also authenticated our results. (Carbohydr. Polym., 2018, 186, 384-393.) He et al. fabricated the fish skin gelatin/TiO2 nanocomposite films. They observed that compound with nano-TiO2 could decrease the light transmittance of gelatin, however, could effectively prevented the UV light which demonstrated our results. (Int. J. Biol. Macromol., 2016, 84, 153-160.)

I commend the authors to include some typical figures for the contact angles measured.

Re: Added. Thanks.

You should change the topic of section 3 to result and discussionsrather than ‘results’. 

Re: Corrected. Thanks.

Section 3.6 I would write the topic in full rather than abreviations ‘’WVP’’

Re: Done. Thanks.

In figure 5 the you can treat the spectrum of TiO2 some correction to make it horizontal like the other spectrum.

Re: Yes, the new spectrum of TiO2 was replaced. Thanks.

Check the first sentence of the new paragraph in the line 332-333. 

Re: We corrected the mistake in the line 332-333. Thanks.

Reviewer 2 Report

The manuscript “Physico-mechanical and antibacterial properties of PLA/TiO2 composite materials synthesized via electrospinning and solution casting processes” by S.Feng and colleagues reports the synthesis and characterization of PLA/TiO2 films and nanofibers. The topic is not new and many articles have been already reported in the literature with similar (but obviously not the same) content. Nevertheless, the article is well-written and organized although some English and editing errors are present. The Introduction and the Materials and Methods sections are well presented and detailed. Honestly, I do not understand why the authors did not perform any statistical test on most of the data presented as bas graph (Fig. 4, 6, 7) as they did on the data presented as table (Tables 1-5) or in Fig. 8 in the Results section. However, in my opinion the manuscript, despite not being a breakthrough article, deserves publication in Coatings.

Author Response

Reviewer #2: Comments:

Honestly, I do not understand why the authors did not perform any statistical test on most of the data presented as bas graph (Fig. 4, 6, 7) as they did on the data presented as table (Tables 1-5) or in Fig. 8 in the Results section.

Re: Added. I have performed the statistical test on all the data presented as bas graph (Fig. 4, 6, 7) as required. All the analyses are to be performed taking three replicates and data will report as mean ± SD. Tests of significant differences between means are to be determined by Duncan’s multiple range tests at a significance level of 0.05 using statistical package for the social sciences SPSS 22. (Ultrasoni. Sonoche., 2017, 36, 11-19.) . Thanks.